# Spatial-Temporal Mutual Distillation for Lightweight Sleep Stage Classification

## Abstract

Sleep stage classification has important clinical significance for the diagnosis of sleep-related diseases. To pursue more accurate sleep stage classification, multi-channel sleep signals are widely used due to the rich spatial-temporal information contained. However, it leads to a great increment in the size and computational costs which constrain the application of multi-channel sleep stage classification models. Knowledge distillation is an effective way to compress models. But existing knowledge distillation methods cannot fully extract and transfer the spatial-temporal knowledge in the multi-channel sleep signals. To solve the problem, we propose a spatial-temporal mutual distillation for multi-channel sleep stage classification. It extracts the spatial-temporal knowledge to help the lightweight student model learn the spatial relationship of human body and the transition rules between multiple sleep epochs. Moreover, the mutual distillation framework improve the teacher by the student model, thus further improve the student model. The results on the ISRUC-III and MASS-SS3 datasets show that our proposed method compresses the sleep models effectively with minimal performance loss and achieves the state-of-the-art performance compared to the baseline methods.

## 1 Introduction

Sleep stage classification plays a crucial role in diagnosing sleep disorders. Sensors are attached to the human body to acquire a set of signals, which is called polysomnography (PSG), including electroencephalography (EEG), electrooculography (EOG), electromyography (EMG). PSG signals are sliced into 30-second segments and then assigned with a sleep stage by human experts following American Academy of Sleep Medicine (AASM) rules (Berry et al., 2012). In AASM rules, five sleep stages are identified: Wake (W), Rapid Eye Movements (REM), Non REM1 (N1), Non REM2 (N2), and Non REM3 (N3) also known as slow wave sleep or even deep sleep. Recently, neural networks are introduced to sleep stage classification to reduce the cost of time and human labor in manual ways.

For accurate sleep stage classification, temporal knowledge is widely used in automatic sleep stage classification. During sleep, the human brain undergoes a series of changes among different sleep stages. For example, the N1 stage often serves as a transition stage between the W stage and other stages. These transition rules are referred as temporal knowledge which are strong references to identify these stages. To capture temporal knowledge, models such as SeqSleepNet (Phan et al., 2019) employ bidirectional Long Short-Term Memory modules.

To meet the higher demands in clinical scenarios, rather than classify with single channel sleep signals, the classification can be greatly improved by utilizing multi-channel sleep signals. Multi-channel sleep signals contain the spatial knowledge which refers to the relationship of the human body. EEG signals, for instance, reflect the structural and functional correlation within the human brain. Models such as those introduced by Andreotti et al. (2018) and Pei et al. (2022) focus on automatically learning the spatial knowledge within multi-channel signals.

However, in the pursuit of automatic sleep stage classification, the size and computational complexity of deep neural networks rapidly increase. This impedes their application in resource-constrained environments, such as clinical care settings or embedded systems. Knowledge distillation is a useful approach to compress neural networks. The compression is realized by transferring the knowledge

from a complex model (teacher model) to a simpler model (student model). However, current knowledge distillation approaches cannot directly be applied to sleep models because of two challenges.

**On the one hand**, existing knowledge distillation approaches cannot fully extract the spatial-temporal knowledge within the multi-channel sleep signals. For example, Liang et al. (2023) introduce multi-level knowledge distillation with a teacher assistant module. Zhang et al. (2022) extract the epoch-wise and sequence-wise knowledge for the distillation. These works only focus on the temporal knowledge but neglect the spatial knowledge. Common knowledge distillation methods like Fitnets (Romero et al., 2014) and Hinton's knowledge distillation (Hinton et al., 2015) cannot closely bound up with the characteristic of sleep signals, which hinders the extraction of sleep spatial-temporal knowledge.

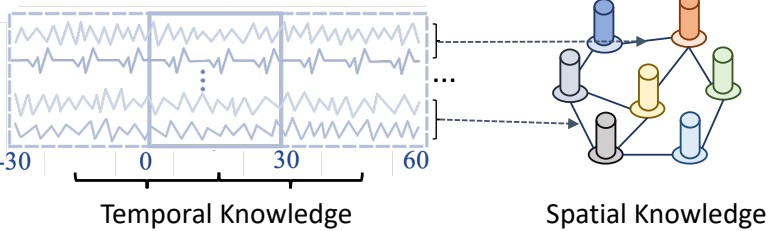

Figure 1: Temporal knowledge represents the transition rules between sleep epochs within a sleep signal sequence. Spatial knowledge means the spatial relationship of multi-channel signals.

**On the other hand**, current knowledge distillation frameworks cannot fully transfer the knowledge to the student model. For example, Hinton et al. (2015); Romero et al. (2014); Zhao et al. (2022); Park et al. (2019) train the teacher model in advance and then distill knowledge to the student model by a fully-trained teacher model which is static in the student training process. They neglect the response of student model to improve the teacher during the student's training procedure, which can in the end improve the student. It constrains the transfer of the spatial-temporal knowledge and the performance of the student model.

To solve the challenges above, we propose a general knowledge distillation framework for multi-channel sleep stage classification models, which can compress the sleep models effectively with minimal performance loss. Our main contributions are as follows:

1. We propose a spatial-temporal knowledge module to fully extract spatial-temporal knowledge from multi-channel sleep signals;

2. We design a mutual distillation framework to improve the transfer of spatial-temporal knowledge.

3. The experiment results indicate that our proposed knowledge distillation framework achieves state-of-the-art performance with two popular architectures, CNN-RNN and CNN-GCN, on both ISRUC-III and MASS-SS3 datasets. It effectively reduces the number of parameters and computational costs of sleep models while preserving its sleep stage classification performance.

## 2 RELATED WORKS

### 2.1 SLEEP STAGE CLASSIFICATION

Sleep stage classification can help diagnose sleep disorders. In earlier studies, researchers employ machine learning methods to classify sleep stages (Tzimourta et al., 2018; Basha et al., 2021; Sundararajan et al., 2021). However, these methods require a large amount of a priori knowledge, which means that a significant manual cost is required to extract features. Therefore, many researchers start to use deep learning methods to extract spatial-temporal knowledge of sleep signals to achieve automatic sleep stage classification.

**For temporal knowledge**, researchers classify sleep stages by capturing contextual dependencies between sleep stages. Based on this, researchers propose a series of sleep stage classification models

that extract temporal knowledge of sleep signals. For example, DeepSleepNet (Supratak et al., 2017) uses Bi-LSTM to extract sequential features of sleep signals; A CNN-based model proposed by Sun et al. (2019) devise a hierarchical neural network to learn temporal features for the sleep stage classification; SleepEEGNet (Mousavi et al., 2019) employs a bidirectional recurrent neural network to capture long-term and short-term contextual dependencies. Both MLP and LSTM are applied by Dong et al. (2017) for the extraction and mining of temporal features.

**For spatial knowledge**, researchers classify sleep stages with multi-channel sleep signals from sensors in different body parts (Gao & Ji, 2019). For example, Chambon et al. (2018) use convolutional layers across channels to extract spatial knowledge. Shi et al. (2015) use the joint collaborative representation to fuse EEG representations and extract spatial knowledge. 2D CNN is applied by Sokolovsky et al. (2019) to capture the spatial knowledge of EEG and EOG. Jia et al. (2023) improve the classification performance of sleep stage classification models by exploring the correlation of individual channels. In addition, there are also methods that extract both temporal relationship and spatial knowledge. For example, MSTGCN (Jia et al., 2021) uses deep graph neural networks to model spatial knowledge for more accurate sleep stage classification of multi-channel sleep signals.

Although these methods achieve good performance in the field of sleep stage classification, the size of the models is rapidly growing. This leads to high computational and storage costs for the models in practical applications, making it difficult to achieve deployment in hardware devices. We introduce the extraction of spatial-temporal knowledge of sleep signals into knowledge distillation to achieve lightweight sleep stage classification.

## 2.2 KNOWLEDGE DISTILLATION

Knowledge distillation is an important approach in model compression. It has two main challenges: knowledge extraction and knowledge transfer.

**For knowledge extraction**, researchers extract knowledge from teacher in multiple ways. In the beginning, Hinton et al. (2015) use the output of the teacher model as a kind of soft label to participate in the training of the student model. For more efficient extraction, new knowledge extraction techniques arise. Fitnets (Romero et al., 2014), for example, use the middle layer features of teacher models as hints to guide student models for training. Park et al. (2019) focus on the multivariate relationship between each sample and transfer the relationship matrix as a kind of knowledge to the student model; Tian et al. (2019) encourage the positive samples to be closer and penalize the negative samples to make them farther away by the relationship between positive and negative samples. Minami et al. (2020) construct relationships as graphs for relationship-based graph knowledge transfer. In conclusion, efficient feature extraction is key to the knowledge distillation.

**For knowledge transfer**, more efficient distillation frameworks are proposed to better transfer knowledge. For example, Mirzadeh et al. (2020) introduces a teaching assistant model to help reduce the gap between teachers and students; Recently, a new type of distillation utilizing mutual learning to help knowledge transfer. In this circumstance, the knowledge is mutually transferred between multiple models. For example, Zhang et al. (2018) abandon the traditional teacher-student architecture and allowed each pair of models in the model set to learn from each other; Ren et al. (2021) introduce a Master to update teacher and student models alternately.

In sleep stage classification task, it is vital to propose a knowledge distillation approach tightly combined with the characteristics of sleep signals. The knowledge distillation approach we proposed extracts the spatial-temporal knowledge of sleep signals and transfers sleep spatial-temporal knowledge in a mutual distillation framework.

## 3 PRELIMINARY

**Definition 1** In the task of sleep stage classification with multi-channel sleep signals, we define the input signals $X$ as follows:

$$X = \begin{bmatrix} x_{11} & \cdots & x_{1L} \\ \vdots & \ddots & \vdots \\ x_{C1} & \cdots & x_{CL} \end{bmatrix}, \quad x_{ij} \in \mathbb{R}^n \tag{1}$$

where, $L$ denotes the length of a sequence. $C$ denotes the number of channels. $n$ represents the length of a sleep epoch.

**Definition 2** Consider a multi-channel sleep stage classification model $f$, which can be represented as the composition $f = f_1 \odot f_2$, where $f_1$ represents the encoder part of the model and $f_2$ represents the classifier.

**Definition 3** The multi-channel features can be obtained by applying $X$ to $f_1$:

$$H = f_1(X) = \begin{bmatrix} h_{11} & \cdots & h_{1L} \\ \vdots & \ddots & \vdots \\ h_{C1} & \cdots & h_{CL} \end{bmatrix}, \quad h_{ij} \in \mathbb{R}^m \tag{2}$$

where, $m$ represents the feature length of an epoch.

**Definition 4** The classification results can be obtained by inputting the feature matrix $H$ into the classifier $f_2$:

$$\hat{Y} = f_2(H) = \{\hat{y}_1, \cdots, \hat{y}_L\} \tag{3}$$
$$\hat{y}_i = \{p_1, \cdots, p_k\}, p_i \in (0, 1) \tag{4}$$

where, $\hat{y}_i$ represents the probability distribution of each class with a length of $k$ and $p_i$ is the probability of the $i$-th class. $k$ corresponds to the number of classes, which is 5 under the AASM standard.

## 4 SPATIAL-TEMPORAL MUTUAL DISTILLATION

As shown in Figure 2, Spatial-Temporal Mutual Distillation consists of the spatial knowledge module, the temporal knowledge module and a mutual distillation framework to compress sleep stage classification models. Specifically, multi-channel features of sleep signals are extracted by the encoder at the beginning. To model the spatial knowledge, a sleep graph is constructed from the multi-channel features to transfer the knowledge from the teacher to the student. As for temporal knowledge, it is modeled by measuring the temporal relationship vector within the sleep signal sequence, thus guiding the student model to learn the temporal knowledge contained in the teacher model. In addition, a mutual distillation framework is designed to further improve the distillation with mutual knowledge transfer.

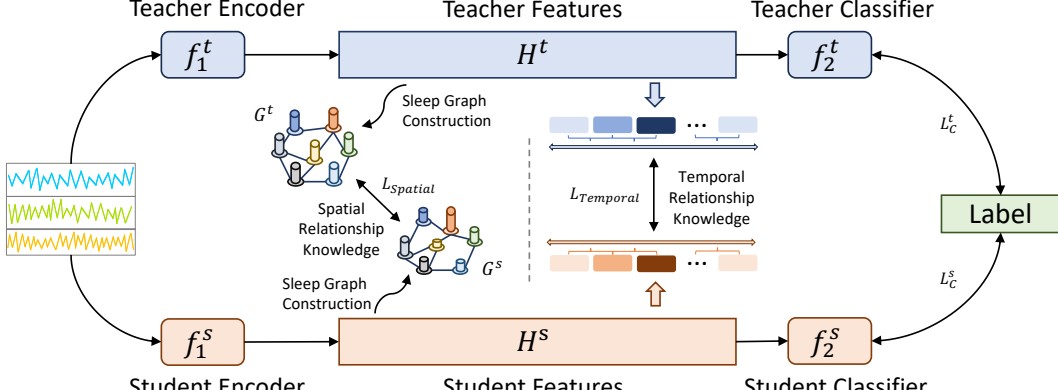

Figure 2: Here is the overall process of Spatial-Temporal Mutual Distillation. Initially, the multi-channel sleep signals are encoded by both the teacher and student encoder, extracting corresponding multi-channel features. Subsequently, the temporal knowledge module and the spatial knowledge module extract spatial-temporal knowledge and then mutually transferred under the mutual distillation framework.

### 4.1 Spatial Knowledge Module

For the extraction spatial knowledge, we design the spatial knowledge module. It starts with sleep graph construction to represent spatial knowledge as a graph. Then, we measure the difference between graphs from the teacher and the student to convey the spatial knowledge.

It is a key question that how to represent the spatial knowledge for better knowledge transfer. Since the spatial knowledge expresses the spatial relationship of multi-channel sleep signals, we construct sleep graph $G = \{V, E\}$ whose edges show the relationship between channels. Sleep graph is constructed from multi-channel features encoded from the multi-channel sleep signals. Each channel can be denoted as a node $v_i$, while the edge between $v_i$ and $v_j$ are denoted as $e_{ij}$. The edge is measured by a regularized form as follows:

$$e_{ij} = \frac{e^{R_s(v_i, v_j)}}{\sum_j^C e^{R_s(v_i, v_j)}} \tag{5}$$

where $R_s$ is cosine similarity function which measures the relationship of each pair of nodes.

In the process of knowledge distillation, knowledge transfer is conducted by utilizing the distance of the teacher and student model. For the sleep graph we propose, we measure the spatial relationship distance of the sleep graphs by bringing the KL divergence to each node. As for node $i$, the spatial relationship distance $D_i$ is calculated as follows:

$$D_i = KLD(e_i^s \| e_i^t) = \sum_{j=0}^C e_{ij}^s \log \frac{e_{ij}^s}{e_{ij}^t} \tag{6}$$

where $e_i^s = \{e_{i1}^s, \cdots, e_{iC}^s\}$ is the spatial relationship vector corresponding to node $v_i^s$ in the student's sleep graph, and $e_i^t = \{e_{i1}^t, \ldots, e_{iC}^t\}$ is the spatial relationship vector corresponding to node $v_i^t$ of the teacher's sleep graph. In this calculation, if $D_i$ is smaller, it means that the teacher's node $v_i^t$ and the student's node $v_i^s$ have more similar spatial knowledge. Therefore, by bringing all nodes into the calculation, the loss function for spatial knowledge can be derived as follows:

$$L_{spatial} = \frac{1}{C} \sum_{i=1}^C D_i \tag{7}$$

### 4.2 Temporal Knowledge Module

Sleep signal sequences naturally contain temporal knowledge. It represents contextual dependencies between epochs. The classification of a certain epoch can be inferred from the relationship with the back-and-forth epochs. In the existing distillation for sleep models, they directly align the features of a sequence instead of modeling the relationship of epochs. It is not accurate and has problems like dimension alignment. To extract the relationship of a sleep signal sequence, we design a temporal knowledge module. We choose to model the relationship between the epochs as contextual constraints over a sequence, which is more in line with the characteristics of sleep signals.

Specifically, the temporal knowledge module is computed as follows: Given sleep signal sequence features $H$ with a length of $L$ epochs. We take $i$-th epoch which contains $C$ channels as $u_i$. To model the relationship between the features $u_i$ and $u_j$ of two epochs, it can be expressed as follows:

$$R_{ij} = R_t(u_i, u_j) \tag{8}$$

where $R_t(\cdot, \cdot)$ denoted a relationship function computed by Euclidean distance. By applying the relationship to all the epochs in the sequence in pairs, we can get a temporal relationship vector $vec = \{R_{ij} | i, j \in [1, L]\}$. After computing the temporal relationship vector for both the teacher and the student model, we can get the corresponding temporal relationship vector denoted as $vec^t$ and $vec^s$. To transfer the temporal knowledge to the student, we calculate the difference of temporal relationship vectors of the teacher and student using the $SmoothL1(\cdot, \cdot)$ loss function, which can be

expressed as follows:

$$L_{temporal} = SmoothL1(vec^t, vec^s) = \begin{cases} 0.5|vec^t - vec^s|^2, |vec^t - vec^s| < 1 \\ |vec^t - vec^s| - 0.5, otherwise \end{cases} \tag{9}$$

### 4.3 MUTUAL DISTILLATION FRAMEWORK

Traditional knowledge distillation employs a static teacher model in the distillation. In the sleep stage classification task, it constrains the knowledge transfer and limits the student's performance. For better knowledge transfer, we design a mutual distillation framework to transfer spatial-temporal knowledge.

On the training epoch $i$, the update of both teacher and student model can be expressed as follows:

$$L_c^t = CE(f_i^t(x), y) = -\sum_i y \cdot \log(f_i^t(x)) \tag{10}$$

$$L_c^s = CE(f_i^s(x), y) = -\sum_i y \cdot \log(f_i^s(x)) \tag{11}$$

$$Loss_i^t = \alpha L_c^t + \beta L_{spatial} + \gamma L_{temporal} \tag{12}$$

$$Loss_i^s = \alpha L_c^s + \beta L_{spatial} + \gamma L_{temporal} \tag{13}$$

where $\alpha$, $\beta$ and $\gamma$ are three hyperparameters which stands for the weights to balance the losses. $L_c^t$ denotes the classification loss of the teacher model while $L_c^s$ denotes the classification loss of the student model.

## 5 EXPERIMENTS

### 5.1 DATASETS

We conduct experiments on two publicly available sleep datasets. These two datasets contain adequate multi-channel signals and are scored by experts according to the AASM manual that can be used for evaluating sleep model performance.

**ISRUC-III** is obtained from a sample of 8,549 PSG sleeps over 8 hours from 10 healthy adult subjects, including one male and nine females. We use 8 subjects as the training set, 1 subject as the validation set, and 1 subject as the test set.

**MASS-SS3** contains 59,056 PSG sleep samples from the sleep data of 62 healthy subjects, including 28 males and 34 females. We also use 50 subjects as the training set, 6 subjects as the validation set, and 6 subjects as the test set.

### 5.2 EXPERIMENT SETTINGS

To conduct a fair comparison, we bring the same data and model settings to all knowledge distillation baselines and our framework. The detail of the baseline methods are shown in the Appendix A.1.

With a sampling rate of 100 Hz for both ISRUC-III and MASS-SS3, The experiments utilize three channels sets of 6-channel EEG/EOG, 8-channel EEG/EOG, and 6-channel EEG. The results of 8-channel EEG/EOG and 6-channel EEG are presented in Appendix A.3 and A.4.

The spatial-temporal knowledge naturally exists in most of the sleep models, whose most popular architecture is CNN-RNN and CNN-GCN. Based on the inspiration of classical sleep models such as CNN-RNN-based TinySleepNet and CNN-GCN-based GraphSleepNet, we design two pairs of multi-channel teacher-student models for the comparison of knowledge distillation frameworks. In the CNN-RNN architecture, we delete the units of the dense layer and the LSTM as well as the number of filters in the CNNs. In the CNN-GCN architecture, we delete units of the Graph Convolution layer and the number of filters in the CNNs. The hyperparameters of the compressed layers are shown in Table 1. Details about the implementation of models are shown in Appendix A.5.

Table 1: The hyperparameters of the models related to the compression.

| Model | Conv Filters | LSTM Units | Dense | Graph Units |
|---|---|---|---|---|
| CNN-RNN Teacher | 128 | 128 | 1024 | / |
| CNN-RNN Student | 32 | 32 | 128 | / |
| CNN-GCN Teacher | 128 | / | 1024 | 1024 |
| CNN-GCN Student | 32 | / | 128 | 32 |

In the implementation of the models, we use RTX 3090 GPU, and TensorFlow 2.9.0 as the deep learning framework. In this paper, we use Adam as the optimizer for each model with a learning rate of 0.0001 and a batch size of 8 during training. We choose cosine similarity as $R_s$ and L2 normalization as $R_t$. We use a weight setting of $\alpha$:$\beta$:$\gamma$ = 1:5:1, and the loss weights of other baseline methods are shown in Appendix A.6.

## 5.3 OVERALL RESULTS

As the experiment results shown in Table 2, the student model demonstrates a remarkable compression on the number of parameters, size, and FLOPS. It shows that the student model distilled by our method reduces both the scale and computational costs. However, the accuracy and F1-score still maintain a performance near the teacher model.

Table 2: The performance, scale and computational costs of the teacher and student model. #Param denotes the number of parameters. Size denotes the storage the model occupied. FLOPS (Floating point Operations Per Second) denotes the computational costs of the model.

| Model | Accuracy | F1-score | #Param | Size | FLOPS |
|---|---|---|---|---|---|
| CNN-RNN Teacher | 83.47% | 80.50% | 8.72M | 34.9MB | 11.34B |
| CNN-RNN Student | 82.42% | 80.06% | 0.29M | 1.2MB | 1.15B |
| CNN-GCN Teacher | 85.93% | 83.95% | 5.49M | 22MB | 1.61B |
| CNN-GCN Student | 84.26% | 81.12% | 2.13M | 8.6MB | 0.034B |

**Knowledge Extraction.** To demonstrate that the efficient extraction of spatial-temporal knowledge, we compare our framework with the baselines without mutual distillation. From the results in Table 4, it can be concluded that our method achieve better performance than all the baselines without mutual distillation. The reason is that our proposed method utilizing spatial-temporal knowledge in the multi-channel sleep signals, while the baselines without mutual distillation only consider incomplete knowledge. For example, Knowledge Distillation and Decoupled Knowledge Distillation only apply the knowledge from the output. Fitnets and Neuron Selectivity Transfer consider the intermediate features but ignore the spatial relationship in multi-channel sleep signals. Relational Knowledge Distillation takes the relationship of contextual epochs into consideration while Distilling Knowledge from GCN models the spatial relationship. Neither of these two approaches takes full account of the spatial-temporal knowledge. The results verify that the spatial-temporal knowledge used by our method is the knowledge should be extracted from the multi-channel sleep signals.

**Knowledge Transfer.** To demonstrate that our knowledge distillation framework can fully transfer the knowledge from the teacher model, we compare with Deep Mutual Learning, a mutual distillation-based framework. Our knowledge distillation framework outperforms Deep Mutual Learning which indicates the strong knowledge transfer ability of our knowledge distillation framework.

## 5.4 VISUALIZATION

Spatial knowledge denotes the functional connectivity of the human body and temporal knowledge denotes the contextual relationship of a sleep sequence. Some researches have shown that the spatial and temporal knowledge varies during different sleep stages. In order to analyze the effectiveness of

Table 3: Comparison with baseline methods on CNN-RNN architecture.

| Method | ISRUC-III | | MASS-SS3 | |
|---|---|---|---|---|
| | Accuracy | F1-score | Accuracy | F1-score |
| Knowledge Distillation | 77.47% | 73.82% | 81.27% | 69.27% |
| Decoupled Knowledge Distillation | 79.26% | 75.68% | 82.51% | 70.61% |
| Fitnets | 78.21% | 73.92% | 81.09% | 67.80% |
| Neuron Selectivity Transfer | 78.42% | 74.18% | 81.79% | 70.04% |
| Relational Knowledge Distillation | 79.26% | 76.75% | 82.55% | 71.30% |
| Distilling Knowledge from GCN | 77.16% | 73.75% | 82.95% | 72.29% |
| Deep Mutual Learning | 80.63% | 77.31% | 82.20% | 70.59% |
| **Ours** | **82.42%** | **80.06%** | **84.22%** | **73.94%** |

Table 4: Comparison with baseline methods on CNN-GCN architecture.

| Method | ISRUC-III | | MASS-SS3 | |
|---|---|---|---|---|
| | Accuracy | F1-score | Accuracy | F1-score |
| Knowledge Distillation | 75.07% | 72.35% | 84.75% | 75.60% |
| Decoupled Knowledge Distillation | 82.44% | 80.26% | 84.79% | 80.32% |
| Fitnets | 81.88% | 80.76% | 84.96% | 75.82% |
| Neuron Selectivity Transfer | 83.31% | 80.94% | 85.51% | 76.81% |
| Relational Knowledge Distillation | 76.68% | 73.19% | 80.5% | 64.19% |
| Distilling Knowledge from GCN | 82.65% | 79.69% | 83.67% | 82.48% |
| Deep Mutual Learning | 81.27% | 77.84% | 83.89% | 72.64% |
| **Ours** | **84.26%** | **81.22%** | **85.71%** | **77.98%** |

spatial and temporal knowledge module, we visualize the sleep graphs constructed by multi-channel sleep signals and temporal relationship at different stages. The results are shown in Figure 3 and Figure 4. It can be summarized from the figures that the sleep graphs and temporal relationship are similar under each sleep period which indicate that our framework transfer the spatial and temporal knowledge efficiently.

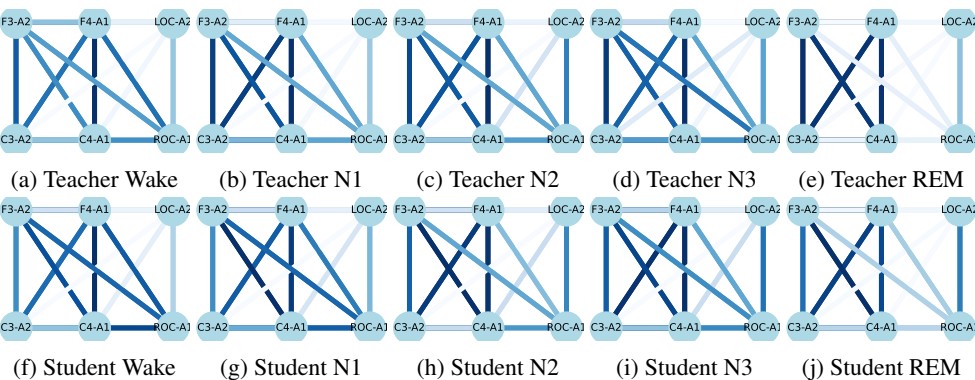

(a) Teacher Wake (b) Teacher N1 (c) Teacher N2 (d) Teacher N3 (e) Teacher REM

(f) Student Wake (g) Student N1 (h) Student N2 (i) Student N3 (j) Student REM

Figure 3: Visualization analysis of spatial knowledge transfer.

## 5.5 ABLATION STUDY

Our method consists of three parts: temporal knowledge module, spatial knowledge module, and mutual distillation framework. This combination forms the optimal performance of spatial-temporal mutual distillation. In order to further study the effectiveness of the method, we conduct ablation experiments to evaluate each specific module and prove the effectiveness of each component of the method.

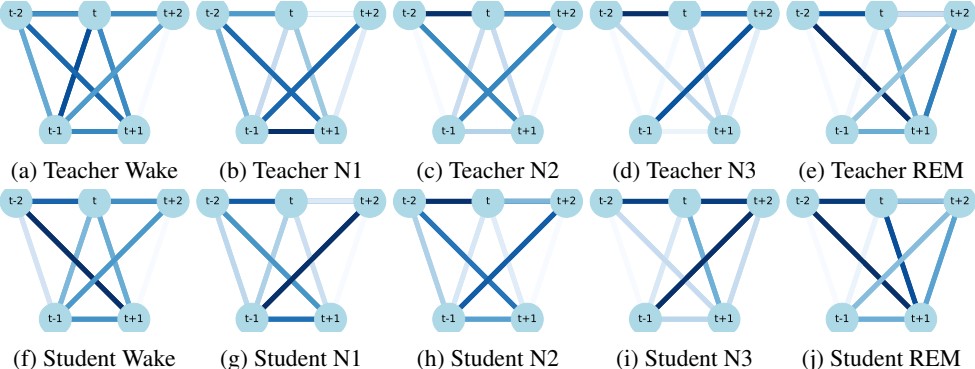

| (a) Teacher Wake | (b) Teacher N1 | (c) Teacher N2 | (d) Teacher N3 | (e) Teacher REM |

| (f) Student Wake | (g) Student N1 | (h) Student N2 | (i) Student N3 | (j) Student REM |

Figure 4: Visualization analysis of temporal knowledge transfer.

The experiment settings of the ablation study are as follows:

- Variant I: Training without neither spatial-temporal knowledge nor mutual distillation framework;
- Variant II: Training with only temporal knowledge module, without mutual distillation framework and spatial knowledge module;
- Variant III: Training with temporal knowledge module and spatial knowledge module, without mutual distillation framework;
- Variant IV: Training with mutual spatial-temporal knowledge distillation.

Through the results shown in Table 5, it can be observed that the temporal knowledge module has a positive impact on the knowledge distillation performance because of transferring the temporal knowledge. Then, the spatial knowledge module also contributes to the performance by extracting and conveying spatial knowledge of multi-channel sleep signals. In addition, the gain of the mutual distillation framework indicates mutual knowledge transfer helps further improve the distillation.

Table 5: The results of each variant.

| Method | ISRUC-III | | MASS-SS3 | |
|---|---|---|---|---|
| | Accuracy | F1-score | Accuracy | F1-score |
| Variant I | 77.47% | 73.82% | 81.27% | 69.27% |
| Variant II | 78.10% | 75.07% | 83.03% | 72.29% |
| Variant III | 80.52% | 77.40% | 83.58% | 73.29% |
| **Ours** | **82.42%** | **80.06%** | **84.22%** | **73.94%** |

## 6 CONCLUSION

We propose a novel knowledge distillation approach for the sleep stage classification task with multi-channel sleep signals. It consists of three parts: spatial knowledge module, temporal knowledge module, and mutual distillation framework. The spatial knowledge module constructs the sleep graph and conveys the spatial knowledge extracted from multi-channel sleep signals. Meanwhile, the temporal knowledge module transfers the relationship between sleep epochs inside a sequence. To further improve the distillation, the mutual distillation framework is designed to mutually transfer the spatial-temporal knowledge between the teacher and student. Our experiments indicate that our method significantly compresses the model while maintaining its performance. It attains state-of-the-art performance on two public sleep datasets, ISRUC-III and MASS-SS3. Furthermore, each component of our method is confirmed effective through the ablation study. The proposed method is a general distillation framework for multi-channel time series classification. In the future, we can apply the proposed method to other large-scale multi-variate time series models.

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

# A  APPENDIX

## A.1  BASELINE METHODS

To evaluate our method, we compare it with multiple baseline knowledge distillation methods on both sleep datasets:

- **Knowledge Distillation (Hinton et al., 2015):** Use the teacher's output probability distribution to guide student's training process.

- **Fitnets (Romero et al., 2014):** Extend the idea of the traditional knowledge distillation by using both the output of the teacher model and the intermediate representation as a hint to the student.

- **Neuron Selectivity Transfer (Huang & Wang, 2017):** Match the distributions of the neuron selectivity patterns with maximum mean discrepancy between the teacher and student networks.

- **Deep Mutual Learning (Zhang et al., 2018):** The teacher and student collaboratively learn and teach each other throughout the entire training process.

- **Relational Knowledge Distillation (Park et al., 2019):** Use distance and angle distillation loss to penalize the difference in relation structure.

- **Distilling Knowledge from GCN (Yang et al., 2020):** Transfer topological semantics of a pre-trained GCN by a local structure preservation module.

- **Decoupled Knowledge Distillation (Zhao et al., 2022):** Re-express the loss as two parts, target class, and non-target class, which focus on the classification correctness and probability distribution separately.

## A.2  DESCRIPTION OF DATASETS

We evaluate our paper on MASS-SS3 (O'reilly et al., 2014) and ISRUC-III (Khalighi et al., 2016) datasets. The details of these datasets can be seen in Table 6.

Table 6: The description of datasets on MASS and ISRUC-III.

| Dataset | Signal Type | Label | Frequency Rate |
|---|---|---|---|
| MASS | EEG | C3 | 256Hz |
| | | C4 | 256Hz |
| | | Cz | 256Hz |
| | | F3 | 256Hz |
| | | F4 | 256Hz |
| | | F7 | 256Hz |
| | | F8 | 256Hz |
| | | O1 | 256Hz |
| | | O2 | 256Hz |
| | | P3 | 256Hz |
| | | P4 | 256Hz |
| | | Pz | 256Hz |
| | | T3 | 256Hz |
| | | T4 | 256Hz |
| | | T5 | 256Hz |
| | | T6 | 256Hz |
| | | Fp1 | 256Hz |
| | | Fp2 | 256Hz |
| | | Fpz | 256Hz |
| | EOG | / | 256Hz |
| | EMG | / | 256Hz |
| | ECG | / | 512Hz |
| ISRUC-III | EOG | LOC-A2 | 100Hz |
| | | ROC-A1 | 100Hz |
| | EEG | F3-A2 | 200Hz |
| | | C3-A2 | 200Hz |
| | | O1-A2 | 200Hz |
| | | F4-A1 | 200Hz |
| | | C4-A1 | 200Hz |
| | | O2-A1 | 200Hz |
| | Chin-EMG | X1 | 200Hz |
| | ECG | X2 | 200Hz |
| | Leg1-EMG | X3 | 200Hz |
| | Leg2-EMG | X4 | 200Hz |

## A.3 EXPERIMENTS ON 6-CHANNEL EEG OF ISRUC-III

For the further evaluation for our method, we conduct experiments on 6-channel EEG of ISRUC-III. The results are as follows:

Table 7: Comparison with baseline methods on 6-channel EEG of ISRUC-III.

| Method | Accuracy | F1-score |
|---|---|---|
| Knowledge Distillation | 84.84% | 82.15% |
| Decoupled Knowledge Distillation | 84.9% | 82.11% |
| Fitnets | 82.32% | 80.67% |
| Neuron Selectivity Transfer | 83.32% | 80.91% |
| Relational Knowledge Distillation | 85.26% | 82.09% |
| Distilling Knowledge from GCN | 84.35% | 81.96% |
| Deep Mutual Learning | 84.18% | 81.88% |
| **Ours** | **86%** | **82.36%** |

In these experiments, our framework still achieves the state-of-the-art performance.

## A.4    Experiments on 6-channel EEG and 2-channel EOG of ISRUC-III

For the further evaluation for our method, we conduct experiments on 6-channel EEG and 2-channel EOG from ISRUC-III with CNN+GCN architecture. The results are as follows:

Table 8: Comparison with baseline methods on 6-channel EEG and 2-channel EOG of ISRUC-III.

| Method | Accuracy | F1-score |
|---|---|---|
| Knowledge Distillation | 73.80% | 69.81% |
| Decoupled Knowledge Distillation | 77.89% | 74.15% |
| Fitnets | 80.53% | 78.09% |
| Neuron Selectivity Transfer | 80.17% | 77.51% |
| Relational Knowledge Distillation | 71.52% | 67.38% |
| Distilling Knowledge from GCN | 82.40% | 80.31% |
| Deep Mutual Learning | 82.41% | 78.64% |
| **Ours** | **84.51%** | **82.38%** |

In these experiments, our framework still achieves the state-of-the-art performance.

## A.5    Details of the Teacher-Student Network

**CNN-RNN-based network.** We design our network with the hybrid architecture of CNN and RNN. This kind of model is usually made up of two parts. One of them is the feature encoder. This part of the network extracts the features from the epochs of each channel by individual encoders. After the encoding, the multi-channel features are concatenated as the input of the rest of the network. The rest of the network consists of a BiLSTM and a dense layer. BiLSTM is employed to capture the contextual features of several continuous epochs during the transition to improve classification accuracy. We use a dense layer as a classifier to generate the output.

As for this model, we use the strategy that quarters the number of kernels in convolution layers in each CNN stream and the number of units in the BiLSTM layer simultaneously. The details of the implementation of the teacher and student are shown in Table 9, 10, 11, and 12.

Table 9: Details of the teacher encoder.

| Layer | Layer Type | #Filters | Size | Stride | Activation | Mode |
|---|---|---|---|---|---|---|
| 1 | Input | / | / | / | / | / |
| 2 | Convolution 1D | 128 | fs/2 | fs/4 | relu | same |
| 3 | Dropout | / | 0.5 | / | / | / |
| 4 | Maxpooling 1D | / | 8 | 8 | / | / |
| 5 | Convolution 1D | 128 | 8 | 1 | relu | same |
| 6 | Convolution 1D | 128 | 8 | 1 | relu | same |
| 7 | Convolution 1D | 128 | 8 | 1 | relu | same |
| 8 | Maxpooling 1D | / | 4 | 4 | / | / |
| 9 | Dropout | / | 0.5 | / | / | / |

Table 10: Details of the rest of the teacher network.

| Layer | Layer Type | Size | Activation |
|---|---|---|---|
| 1 | Encoder | / | / |
| 2 | Concatenate | / | / |
| 3 | BiLSTM | 128 | / |
| 4 | Dropout | 0.5 | / |
| 5 | Dense | 5 | softmax |

Table 11: Details of the student encoder.

| Layer | Layer Type | #Filters | Size | Stride | Activation | Mode |
|-------|------------|----------|------|--------|------------|------|
| 1 | Input | / | / | / | / | / |
| 2 | Convolution 1D | 32 | fs/2 | fs/4 | relu | same |
| 3 | Dropout | / | 0.5 | / | / | / |
| 4 | Maxpooling 1D | / | 8 | 8 | / | / |
| 5 | Convolution 1D | 32 | 8 | 1 | relu | same |
| 6 | Convolution 1D | 32 | 8 | 1 | relu | same |
| 7 | Convolution 1D | 32 | 8 | 1 | relu | same |
| 8 | Maxpooling 1D | / | 4 | 4 | / | / |
| 9 | Dropout | / | 0.5 | / | / | |

Table 12: Details of the rest of the student network.

| Layer | Layer Type | Size | Activation |
|-------|------------|------|------------|
| 1 | Encoder | / | / |
| 2 | Concatenate | / | / |
| 3 | BiLSTM | 32 | / |
| 4 | Dropout | 0.5 | / |
| 5 | Dense | 5 | softmax |

**CNN-GCN-based Network.** We design a teacher-student network based on CNN-GCN architecture. It is made up of two parts: feature extractor and GCN. The feature extractor extracts the features from the epochs of each channel by individual encoders as the input of GCN. GCN learns the spatial knowledge from the multi-channel sleep signals.

We use the strategy of deleting the number of CNN filters and Graph Convolution units. The feature extractor structure is the same as the encoder of the CNN-RNN-based network. The details of GCN are shown in Table 13 and Table 14.

Table 13: Details of the rest of the teacher network.

| Layer | Layer Type | Size | Activation |
|-------|------------|------|------------|
| 1 | Feature Extractor | / | / |
| 2 | Concatenate | / | / |
| 3 | Graph Convolution | 1024 | / |
| 4 | Dropout | 0.5 | / |
| 5 | Dense | 5 | softmax |

Table 14: Details of the rest of the student network.

| Layer | Layer Type | Size | Activation |
|-------|------------|------|------------|
| 1 | Feature Extractor | / | / |
| 2 | Concatenate | / | / |
| 3 | Graph Convolution | 32 | / |
| 4 | Dropout | 0.5 | / |
| 5 | Dense | 5 | softmax |

## A.6 THE HYPERPARAMETERS FOR TRAINING

Here are the hyperparameters we choose for the experiments:

Table 16: Results under different weights settings.

| Weights($\alpha, \beta, \gamma$) | Accuracy | F1-score |
|---|---|---|
| (1,1,1) | 84.25% | 81.93% |
| (1,3,1) | 84.69% | 81.96% |
| (1,5,1) | 84.26% | 81.22% |
| (1,7,1) | 84.07% | 81.60% |

Table 15: Training parameters for baseline methods.

| Method | Epoch | | loss weights | |
| | ISRUC-III | MASS | ISRUC-III | MASS |
|---|---|---|---|---|
| Knowledge Distillation | 30 | | 0.1:0.9 | |
| Fitnets | 25 | | 0.5:0.5 | |
| Neuron Selectivity Transfer | 25 | | 0.5:0.5 | |
| Relational Knowledge Distillation | 25 | | 0.5:0.5 | |
| Distilling Knowledge from GCN | 25 | | 1:5 | |
| Decoupled Knowledge Distillation | 25 | | 1:1 | |
| Deep Mutual Learning | 25 | | 0.1:0.9 | |

## A.7 Sensitivity of Weights in Knowledge Distillation

The weights of each loss term are important hyperparameters in our knowledge distillation framework. We test several weights under same experiments settings. The results using the CNN-GCN-based architecture on ISRUC-III dataset are shown in Table 16 which indicate that our framework is not sensitive to the weights of each loss term.

## A.8 Different Distance Measurements

In the previous experiments, we employ KL Divergence to measure the distance between two sleep graph. We further test other distance measurement function on the same experiment settings. The results are shown in Figure 17. All measurement function can perform well under our knowledge distillation framework and Wasserstein distance reaches the best performance in this experiment.

Table 17: The results of different measurement functions on ISRUC-III.

| Measurements | Accuracy | F1-score |
|---|---|---|
| KL Divergence | 84.26% | 81.22% |
| MMD | 84.50% | 82.1% |
| Wasserstein distance | 85.20% | 82.82% |

## A.9 Clinical Demands

Although reducing scale and computational costs is important for the application of sleep models, the absolute performance should still meet clinical demands. For clarification of the student model distilled by our method meets the clinical demands, we compare its performance with some other popular sleep models. As shown in Table 18, we can find that our student model's performance is close to other sleep stage classification methods. The compression will not lead to the loss of the value of clinical diagnosis in our method.

## A.10 Discussion and Limitation

In this paper, we propose a knowledge distillation framework to compress the compact sleep models for sleep stage classification. In this framework, the proposed framework conveys spatial knowledge

Table 18: Performance of other popular sleep models and our student model on ISRUC-III. The performances are similar which means that, our student model still meets the clinical demands after compression.

| Model | Accuracy | F1-score |
|---|---|---|
| SleepEEGNet | 78.07% | 69.52% |
| SleepUtime | 78.86% | 72.26% |
| Multi-channel DeepSleepNet | 81.89% | 80.16% |
| Student (Ours) | 82.42% | 80.06% |

and temporal knowledge mutually between the teacher to the student. To the best of our knowledge, this framework can not only be applied in sleep stage classification tasks but also can be broadly applied in other classification tasks multi-channel signals like emotion recognition and motor imagery tasks.

We only consider the application of our method in the sleep stage classification task, which would be one of the limitations of our work. There are many physiological signal processing tasks like motor imagery and emotion recognition. In the future, we will do further research on knowledge distillation applied to other physiological signal processing tasks.

### A.11 SOCIAL IMPACT

The proposed mutual spatial-temporal knowledge distillation method for multi-channel physiological signals has a significant social impact in the diagnosis and treatment of sleep-related diseases. With the increasing use of multi-channel physiological signals for automatic sleep stage classification, the size of the models and computational costs are major constraints. However, our proposed method efficiently compresses the current large-scale sleep stage classification models with the smallest loss of performance. Based on this, the acceleration and high performance of the sleep stage classification models will significantly popularize the diagnosis of sleep disorders, thus making more patients be able to get diagnosis and treatments for sleep-related diseases. Moreover, we are able to employ this method on wearable devices so that sleep stage classification will not be limited in hospitals.

### A.12 CONFUSION MATRICES FOR OUR METHOD

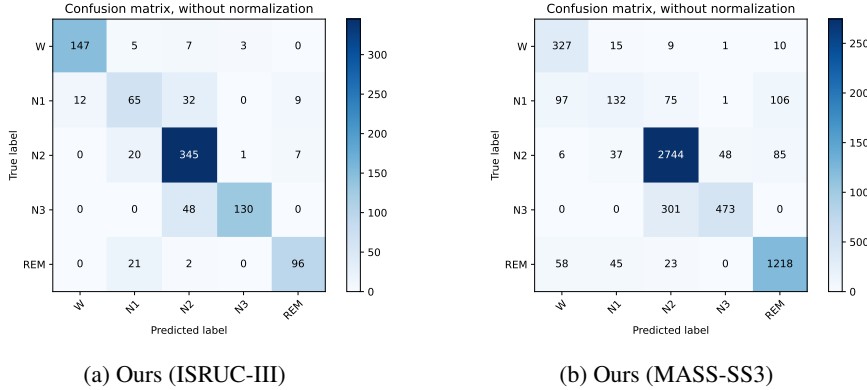

(a) Ours (ISRUC-III)                    (b) Ours (MASS-SS3)

Figure 5: The confusion matrix of the student compressed by the proposed method on the ISRUC-III and MASS-SS3 datasets.

## A.13   CONFUSION MATRICES FOR ABLATION EXPERIMENTS

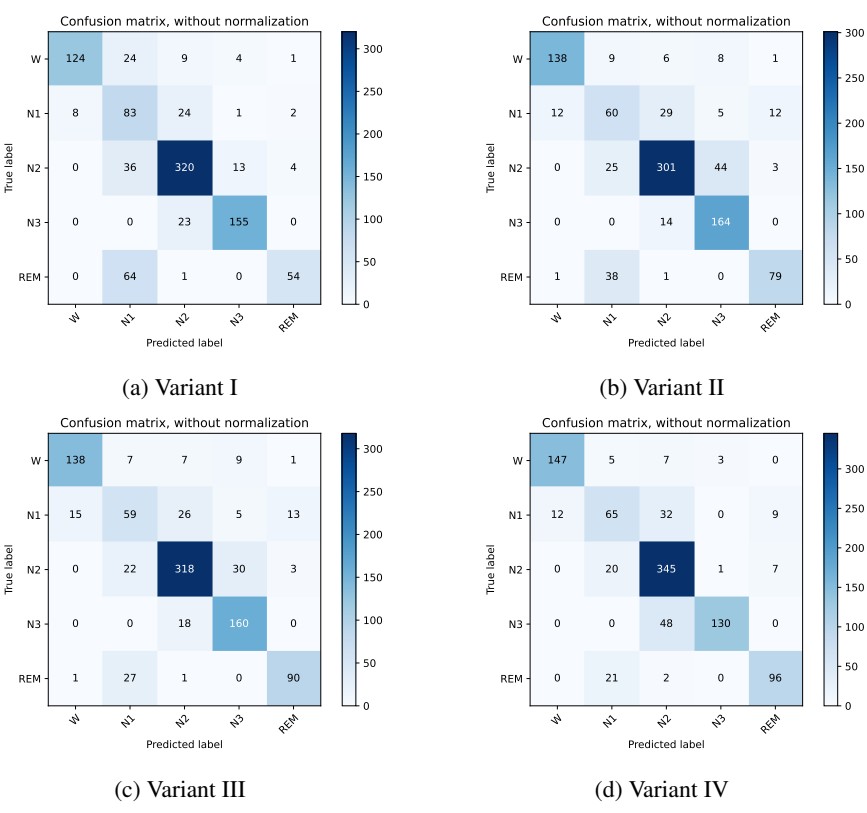

Figure 6: The confusion matrix for the ablation experiments on the ISRUC-III dataset.

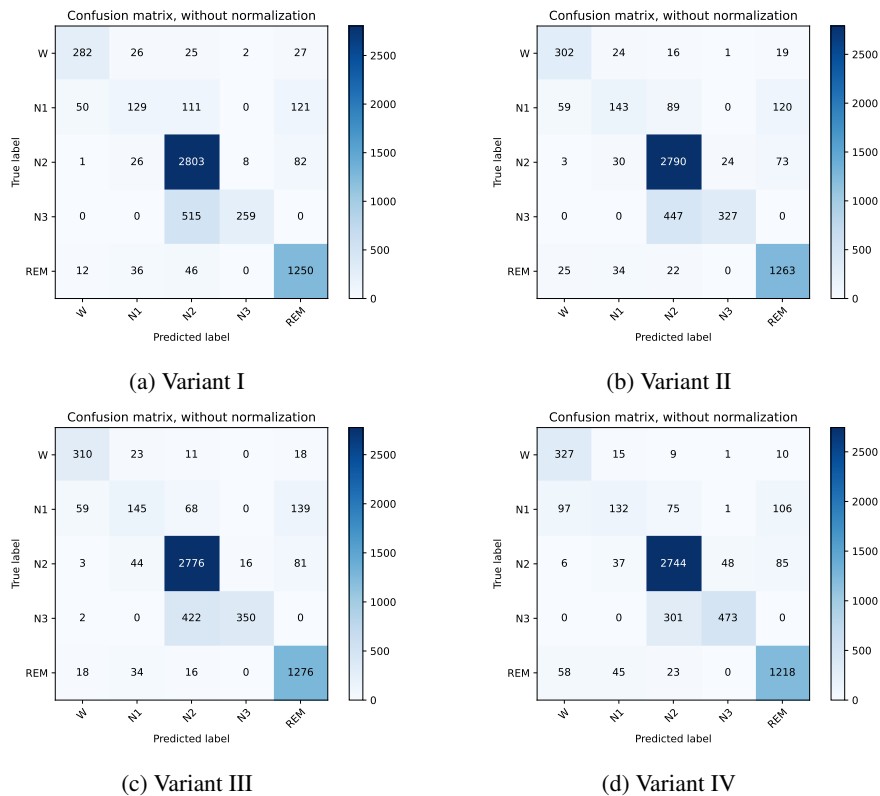

(a) Variant I

(b) Variant II

(c) Variant III

(d) Variant IV

Figure 7: The confusion matrix for the ablation experiments on the MASS-SS3 dataset.

