# OpenReview forum: "Spatial-Temporal Mutual Distillation for Lightweight Sleep Stage Classification"
_ICLR.cc/2024/Conference — ICLR 2024 Conference Withdrawn Submission_

### Official Review · Reviewer_c6gb · 2023-10-27

**Soundness:** 3 good
**Presentation:** 3 good
**Contribution:** 3 good
**Rating:** 8
**Confidence:** 4

**Summary:**

The paper proposes a knowledge distillation mechanism for sleep staging. The authors extract the temporal and spatial features based on multi-channel signals.

Meanwhile, a mutual distillation framework is proposed to update teacher and student model. The method was evaluated on two public datasets.

**Strengths:**

1)The question is interesting. The paper adopts Sleep Knowledge Distillation to decrease the size and computational costs of the existing multi-channel sleep stage
classification models;
2)The paper proposes a spatial-temporal relationship knowledge module to fully extract spatial-temporal knowledge from multi-channel sleep signals
3) This seems to be the first time that spatiotemporal knowledge distillation has been applied to the classification of sleep stages.

**Weaknesses:**

The experiments are not sufficient enough. The authors should compare their KD method with more popular sleep models. In fact, there are some sleep staging model consisting of a less number of model parameters, e.g. TinySleepNet. 3. “As a result, based on the inspiration of classical sleep models such as DeepSleepNet, we design a CNN and RNN based teacher-student model ” . When compared to DeepSleepNet which is a CNN-LSTM based model, TinySleepNet is both more Reviewer #3 lightweight and shares the same structure, so why not design the teacher-student model based on TinySleepNet.

The main purpose of this paper is to achieve student model lightweighting through knowledge distillation. However, as shown in Tab.2, only the computational costs of teacher and student models are listed. It lacks the comparison with other classical sleep staging models.

It lacks the comparison with other multi-channel sleep staging model, such as SalientSleepNet

**Questions:**

See Weaknesses

---

> ### Author Response · Authors · 2023-11-22
> **Response to Reviewer c6gb**
>
> > Q1: The experiments are not sufficient enough. The authors should compare their KD method with more popular sleep models. In fact, there are some sleep staging model consisting of a less number of model parameters, e.g. TinySleepNet. 3.“As a result, based on the inspiration of classical sleep models such as DeepSleepNet, we design a CNN and RNN based teacher-student model ” . When compared to DeepSleepNet which is a CNN-LSTM based model, TinySleepNet is both more lightweight and shares the same structure, so why not design the teacher-student model based on TinySleepNet.
>
> A1: We agree that TinySleepNet is more lightweight and a better choice than DeepSleepNet. DeepSleepNet and TinySleepNet are both CNN-RNN architecture. The design of our CNN-RNN multi-channel teacher-student model is inspired by both DeepSleepNet and TinySleepNet.
>
> In fact, as shown in Appendix A.5, our CNN-RNN model is more similar with TinySleepNet but using multiple channels.
>
> The CNN part of the teacher model:
>
> > | Layer | Layer Type | #Filters | Size | Stride | Activation | Mode |
> > | ----- | ---------- | -------- | ---- | ------ | ---------- | ---- |
> > | 1     | Input      | /        | /    | /      | /          | /    |
> > | 2     | Conv1D     | 128      | fs/2 | fs/4   | Relu       | Same |
> > | 3     | Dropout    | /        | 0.5  | /      | /          | /    |
> > | 4     | Maxpool1D  | /        | /    | 8      | /          | /    |
> > | 5     | Conv1D     | 128      | 8    | 1      | Relu       | Same |
> > | 6     | Conv1D     | 128      | 8    | 1      | Relu       | Same |
> > | 7     | Conv1D     | 128      | 8    | 1      | Relu       | Same |
> > | 8     | Maxpool1D  | /        | 4    | 4      | /          | /    |
> > | 9     | Dropout    | /        | 0.5  | /      | /          | /    |
>
> The RNN part of the teacher model:
>
> > | Layer | Layer Type  | Size | Activation |
> > | ----- | ----------- | ---- | ---------- |
> > | 1     | Encoder     | /    | /          |
> > | 2     | Concatenate | /    | /          |
> > | 3     | BiLSTM      | 128  | /          |
> > | 4     | Dropout     | 0.5  | /          |
> > | 5     | Dense       | 5    | Softmax    |
>
> The CNN part of the student model:
>
> > | Layer | Layer Type | #Filters | Size | Stride | Activation | Mode |
> > | ----- | ---------- | -------- | ---- | ------ | ---------- | ---- |
> > | 1     | Input      | /        | /    | /      | /          | /    |
> > | 2     | Conv1D     | 32       | fs/2 | fs/4   | Relu       | Same |
> > | 3     | Dropout    | /        | 0.5  | /      | /          | /    |
> > | 4     | Maxpool1D  | /        | /    | 8      | /          | /    |
> > | 5     | Conv1D     | 32       | 8    | 1      | Relu       | Same |
> > | 6     | Conv1D     | 32       | 8    | 1      | Relu       | Same |
> > | 7     | Conv1D     | 32       | 8    | 1      | Relu       | Same |
> > | 8     | Maxpool1D  | /        | 4    | 4      | /          | /    |
> > | 9     | Dropout    | /        | 0.5  | /      | /          | /    |
>
> The RNN part of the student model:
>
> > | Layer | Layer Type  | Size | Activation |
> > | ----- | ----------- | ---- | ---------- |
> > | 1     | Encoder     | /    | /          |
> > | 2     | Concatenate | /    | /          |
> > | 3     | BiLSTM      | 32   | /          |
> > | 4     | Dropout     | 0.5  | /          |
> > | 5     | Dense       | 5    | Softmax    |
>
> Furthermore, to further show the generality of our framework, we add new experimentson a CNN+GCN architecture model and add the results to section 5.3 Table 4 :
>
> > | Method                            | Accuracy | F1-score |
> > | --------------------------------- | -------- | -------- |
> > | Knowledge Distillation            | 75.07%   | 72.35%   |
> > | Decoupled Knowledge Distillation  | 82.44%   | 80.26%   |
> > | Fitnets                           | 81.88%   | 80.76%   |
> > | Neuron Selectivity Transfer       | 83.31%   | 80.94%   |
> > | Relational Knowledge Distillation | 76.68%   | 73.19%   |
> > | Distilling Knowledge from GCN     | 82.65%   | 79.69%   |
> > | Deep Mutual Learning              | 81.27%   | 77.84%   |
> > | Ours                              | 84.26%   | 81.22%   |
>
> > Q2: The main purpose of this paper is to achieve student model lightweighting through knowledge distillation. However, as shown in Tab.2, only the computational costs of teacher and student models are listed. It lacks the comparison with other classical sleep staging models.
>
> A2: The method we proposed is a generic knowledge distillation framework that can be applied on most multi-channel sleep models. It is not model-specific. The teacher and student model we use are only a representitive of multi-channel sleep models and their corresponding student models. We compare computational costs of teacher and student models to show the effectiveness of the knowledge distillation framework. Since other classical sleep staging models can be compressed by our knowledge distillation framework, it is meaningless to compare the student model with other models.

---

> > ### Comment · Reviewer_c6gb · 2023-12-02
> > **Feedback**
> >
> > Thank you to the authors for their replies. All my concerns have been resolved. The authors also added additional experiments, which further improved the quality of the paper. Therefore, I maintain the same score as before (Accept).

---

### Official Review · Reviewer_xpJP · 2023-10-27

**Soundness:** 3 good
**Presentation:** 4 excellent
**Contribution:** 3 good
**Rating:** 6
**Confidence:** 3

**Summary:**

Taking into account spatial knowledge in sleep staging can help improve the model prediction's accuracy. Yet increasing the number of channels can also lead to high computational costs. The authors propose a method to reduce the size of the trained model using knowledge distillation. The paper proposes a new way of extracting spatial and temporal knowledge and an effective knowledge transfer. This new approach is tested over sleep data from two different datasets: MASS-SS3 and ISRUC-III.

**Strengths:**

The paper explained the relative work and their method very clearly thanks to good writing and very readable plots.

The authors propose a new way to deal with spatial information and a new transfer strategy between teacher and student called mutual distillation. This new way of transfer allows for a proper exchange of spatial-temporal knowledge between the two models. This new distillation's importance is proved in complete experimental results comprising a comparison between SOTA knowledge distillation methods and ablation study.

**Weaknesses:**

The new mutual distillation needs weight setting. The authors only say that they fixed the weight to 1:5:1 without explaining why this choice was made. The results should be sensitive to the variation of these parameters. Did you try to see the sensitivity of the model to this parameter? How do you select them?

The authors choose only to use six channels comprising EEG and EOG. In classical sleep staging papers, one usually uses 2 EEGs (like Fpz-Cz and Pz-Cz) and possibly EOG (helping to predict the REM stage) and even EMG (helping to predict the wake stage). How was the number of channels chosen? Why six and not all the available channels? The paper shows in the appendix that using the six available EEG channels in ISTUC-III gives better results.

Recent paper, such as Usleep or RobustSleepNet

There is no access to the authors' code for reproducibility; maybe it will be available for the potential final version.

**Questions:**

To characterize the divergence between the two graphs, you chose to use KL divergence. Do you try other divergences such as TV, MMD, or Wasserstein distance?

Do you try to visualize the graph you obtain at the end of the training or even during training? Do we retrieve a graph base of the Euclidean distance between the channels?

Recent architectures, such as Usleep (https://www.nature.com/articles/s41746-021-00440-5) or RobustSleepNet (https://arxiv.org/abs/2101.02452), train their model on several datasets and give good results on an unseen dataset. Distillation learning proposes to have a smaller model for inference, but I am afraid that your student model will be dataset-specific. It will be interesting to see if such a model can be generalized over a new dataset. With maybe more variability in training set (several dataset)

---

> ### Author Response · Authors · 2023-11-22
> **Response to Reviewer xpJP (1/2)**
>
> 1. > The new mutual distillation needs weight setting. The authors only say that they fixed the weight to 1:5:1 without explaining why this choice was made. The results should be sensitive to the variation of these parameters. Did you try to see the sensitivity of the model to this parameter? How do you select them?
>
>    These weights are hyperparameters of the experiments. We select them by experiments. We add more experiments under different weights settings to see the sensitivity of these hyperparameters. We have added the analysis of sensitivity in the Appendix A.7:
>
>    > The weights of each loss term are important hyperparameters in our knowledge distillation frame- work. We test several weights under same experiments settings. The results using the CNN-GCN- based architecture on ISRUC-III dataset are shown in Table 16 which indicate that our framework is not sensitive to the weights of each loss term.
>    >
>    > | Weights | Accuracy | F1-score |
>    > | ------- | -------- | -------- |
>    > | 1:1:1   | 84.25%   | 81.93%   |
>    > | 1:3:1   | 84.69%   | 81.96%   |
>    > | 1:5:1   | 84.26%   | 81.22%   |
>    > | 1:7:1   | 84.07%   | 81.60%   |
>
>    The experiments show that our framework is not sensitive to the variation of these parameters.
>
> 2. > The authors choose only to use six channels comprising EEG and EOG. In classical sleep staging papers, one usually uses 2 EEGs (like Fpz-Cz and Pz-Cz) and possibly EOG (helping to predict the REM stage) and even EMG (helping to predict the wake stage). How was the number of channels chosen? Why six and not all the available channels? The paper shows in the appendix that using the six available EEG channels in ISTUC-III gives better results.
>
>    We choose six channels of EEG and then replace 2 EEG channels with 2 EOG to test the generality under a fixed amount of channels. We follow your advice and conduct experiments on 6 EEG and 2 EOG from ISRUC-III. We have added the experiment results in Appendix A.4:
>
>    > For the further evaluation for our method, we conduct experiments on 6-channel EEG and 2-channel EOG from ISRUC-III with CNN+GCN architecture. The results are as follows:
>    >
>    > | KD Methods                        | Accuracy | F1-score |
>    > | --------------------------------- | -------- | -------- |
>    > | Knowledge Distillation            | 73.80%   | 69.81%   |
>    > | Decoupled Knowledge Distillation  | 77.89%   | 74.51%   |
>    > | Fitnets                           | 80.53%   | 78.09%   |
>    > | Neuron Selectivity Transfer       | 80.17%   | 77.51%   |
>    > | Relational Knowledge Distillation | 71.52%   | 67.38%   |
>    > | Distilling Knowledge from GCN     | 82.40%   | 80.31%   |
>    > | Deep Mutual Learning              | 82.41%   | 78.64%   |
>    > | Ours                              | 84.51%   | 82.38%   |
>    >
>    > In these experiments, our framework still achieves the state-of-the-art performance.The  results are as follows:
>
>    The results show that our framework reaches the state-of-the-art performance and shows good generality on different channels.
>
> 3. > To characterize the divergence between the two graphs, you chose to use KL divergence. Do you try other divergences such as TV, MMD, or Wasserstein distance?
>
>    Thank you for your advice. The KL divergence is one of the many ways to measure the difference of two graphs. It can be changed into any other reasonable measurement methods. We conduct experiments with MMD and Wasserstein distance on the CNN+GCN structure. The results are shown in Appendix A.8:
>
>    > In the previous experiments, we employ KL Divergence to measure the distance between two sleep graph. We further test other distance measurement function on the same experiment settings. The results are shown in Figure 17. All measurement function can perform well under our knowledge distillation framework and Wasserstein distance reaches the best performance in this experiment.
>    >
>    > |  Method  | KL divergence | MMD  | Wasserstein distance |
>    > | :------: | :-----------: | :--: | :------------------: |
>    > | Accuracy |     84.26     | 84.5 |         85.2         |
>    > | F1-score |     81.22     | 82.1 |        82.82         |
>
>    The results show that Wasserstein distance reaches the best result which is declared it in the paper.

---

> ### Author Response · Authors · 2023-11-22
> **Response to Reviewer xpJP (2/2)**
>
> 4. > Do you try to visualize the graph you obtain at the end of the training or even during training? Do we retrieve a graph base of the Euclidean distance between the channels?
>
>    We visualize the spatial and temporal relationship in the figure 3 and 4 in the new manuscript. The student model compressed by our knowledge distillation framwork shows similar spatial and temporal relationship with the teacher model, which proves the effectiveness of our framework on transfering spatial and temporal knowledge.
>
> 5. > Recent architectures, such as Usleep (https://www.nature.com/articles/s41746-021-00440-5 (https://www.nature.com/articles/s41746-021-00440-5)) or RobustSleepNet (https://arxiv.org/abs/2101.02452 (https://arxiv.org/abs/2101.02452)), train their model on several datasets and give good results on an unseen dataset. Distillation learning proposes to have a smaller model for inference, but I am afraid that your student model will be dataset-specific. It will be interesting to see if such a model can be generalized over a new dataset. With maybe more variability in training set (several dataset)
>
>    These two models are excellent models in sleep stage classification area. Our framework can also be used on these two models and make them smaller and easier to apply. We will carry out further researches on applying our framework on these two models.

---

> > ### Comment · Reviewer_xpJP · 2023-12-02
> >
> > I thank the authors for their interesting response

---

### Official Review · Reviewer_kX3p · 2023-11-01

**Soundness:** 3 good
**Presentation:** 3 good
**Contribution:** 2 fair
**Rating:** 5
**Confidence:** 4

**Summary:**

This paper investigates how to **effectively apply** knowledge distillation (KD) in the spatiotemporal sleep-stage classification task, motivated by the requirement of model efficiency.

This paper proposes a specific KD solution **tightly combined with the characteristics of sleep signals** by mainly addressing two unique challenges in this task:
(1) **what knowledge types** related to spatiotemporal signals are useful; and
(2) **how to transfer** the spatial-temporal knowledge.

**Strengths:**

1. This paper studies an interesting and practical research topic: how to efficiently and effectively transfer knowledge for spatiotemporal signals/models?

2. This paper has its novelty--it proposes a specific KD solution tightly combined with the characteristics of sleep signals.

There are two technical contributions:
- **Novel knowledge types for sleep signal:** Channel-to-channel pairwise distances are treated as spatial knowledge type. Epoch-to-epoch pairwise similarities are treated as temporal knowledge type.
- The paper uses a combination of Mutual Distillation [1] and Relational KD [2] to perform the teacher-student knowledge **transfer**.

[1] Zhang, Ying, et al. "Deep mutual learning." Proceedings of the IEEE conference on computer vision and pattern recognition. 2018.
[2] Park, Wonpyo, et al. "Relational knowledge distillation." Proceedings of the IEEE/CVF conference on computer vision and pattern recognition. 2019.

3. The paper conducted comprehensive experiments to justify the effectiveness of the proposed method.

**Weaknesses:**

1. Despite its novelty in the sleep classification domain. The technical contributions of this paper might be relatively incremental in the ML/AI community.

For example, the second contribution ("how to transfer") is just a combination of [1] and [2].

2. The paper has missing discussions on the existing Spatiotemporal Machine Learning works that is not limited to Sleep Analysis field. Spatiotemporal Machine Learning is already a large research area, and a lot of KD-related works emerge in this field.

3. The paper has missing discussions on why the proposed knowledge types are reasonable.

This paper encodes spatial & temporal knowledge in a separate manner. However, is it not true that spatiotemporal knowledge should be jointly modeled as they are hight entangled?

Given this awareness, the proposed knowledge types ("Channel-to-channel distances" and "Epoch-to-epoch similarities") might be outdated, unless the authors can provide the evidence why the separate knowledge types are better than the jointly encoded knowledge.

**Questions:**

In Eq.(11-12), shouldn't the Loss terms are gradient?

Also, Eq.(13-14) never appear in objectives. Where should they belong to?

---

> ### Author Response · Authors · 2023-11-22
> **Response to Reviewer kX3p**
>
> > Q1: Despite its novelty in the sleep classification domain. The technical contributions of this paper might be relatively incremental in the ML/AI community. For example, the second contribution ("how to transfer") is just a combination of [1] and [2].
> >
> > [1] Zhang, Ying, et al."Deep mutual learning." Proceedings of the IEEE conference on computer vision and pattern recognition. 2018.
> >
> > [2] Park, Wonpyo, et al."Relational knowledge distillation." Proceedings of the IEEE/CVF conference on computer vision and pattern recognition. 2019.
>
> A1: We agree that our novelty is limited to sleep stage classification domain, and mutual distillation is widely studied in other area. Nevertheless, it cannot be ignored that the application of sleep models still lacks of exploration and  sleep model compression framework is needed. We root on the spatial and temporal knowledge in sleep stage classification area and adopt knowledge distillation into this area to meet the demand for compression and application in sleep stage classification. As we can see from the experiment results, knowledge distillation frameworks from other area cannot adapt well in the sleep stage classification area. We specifically design the knowledge distillation framework based on the spatial and temporal semantic prior knowledge in the sleep stage classification.
>
> > Q2: The paper has missing discussions on the existing Spatio-temporal Machine Learning works that is not limited to Sleep Analysis field. Spatio-temporal Machine Learning is already a large research area, and a lot of KD-related works emerge in this field.
>
> A2: Thanks for your advice. It is true that Spatio-temporal Machine Learning is widely used in sleep stage classification and knowledge distillation. It can be a great help in extracting and transferring spatial temporal knowledge in the distillation. Our work focuses specifically on compress sleep models with knowledge distillation and we have discussed some spatio-temporal Machine Learning works in sleep stage classification. We will further conduct researches on introducing more advanced spatio-temporal Machine Learning works into our area.
>
> > Q3: The paper has missing discussions on why the proposed knowledge types are reasonable.
>
> A3: The proposed knowledge types are based on the knowledge of sleep stage classification. The spatial knowledge extracted from multi-channel sleep signals represents the functional relationship of human body. The temporal knowledge denotes the contextual relationship over time and the transition rules of multiple sleep epochs. These two kinds of knowledge are widely used and tested in the sleep stage classification area.
>
> To help better understand the spatial-temporal knowledge, we visualize the spatial and temporal connections between multi-channel sleep signals to better demonstrate the transfer of spatial and temporal knowledge. It is shown in the figure.3 and 4 in the new manuscript we upload.
>
> > Q4: This paper encodes spatial & temporal knowledge in a separate manner. However, is it not true that spatiotemporal knowledge should be jointly modeled as they are hight entangled? Given this awareness, the proposed knowledge types ("Channel-to-channel distances" and "Epoch-to-epoch similarities") might be outdated, unless the authors can provide the evidence why the separate knowledge types are better than the jointly encoded knowledge.
>
> A4: It is true that spatial and temporal knowledge is highly entangled. We focus on the explicit extraction and transfer of spatial-temporal knowledge and visualize them to present a more pratical implications, making it easier to understand and explain in the clinical scenerios. As we can see in the figure 3 and 4 in the new manuscript, the spatial and temporal knowledge are visualized as the connections, which is more easier to understand. Despite that, it is free to classify the sleep stage after the fusion of spatial-temporal knowledge in the model we choose to compress.
>
> > Q5: In Eq.(11-12), shouldn't the Loss terms are gradient?
>
> A5: Thank you for your correction. We have corrected them in the new version of manuscript:
>
> > On the training epoch i, the update of both teacher and student model can be expressed as follows:
> > $$
> > L^t_{c} = CE(f^t_i(x), y) = -\sum_{i} y \cdot \log(f^t_i(x))
> > $$
> >
> > $$
> > L^s_{c} = CE(f^s_i(x), y) = -\sum_{i} y \cdot \log(f^s_i(x))
> > $$
> >
> > $$
> > Loss^t_i = \alpha L^t_{c} + \beta L_{spatial} + \gamma L_{temporal}
> > $$
> >
> > $$
> > Loss^s_i = \alpha L^s_{c} + \beta L_{spatial} + \gamma L_{temporal}
> > $$
>
> > Q6: Also, Eq.(13-14) never appear in objectives. Where should they belong to?
>
> A6: Eq.(13-14) respectively calculate the teacher and student's hard loss of the classification, which is the crossentropy of the label and the prediction. They are included in Eq.(11-12) .

---

> > ### Comment · Reviewer_kX3p · 2023-12-03
> > **Feedback to Authors Response**
> >
> > Thank the authors for the reply. I want to keep my scores after reading the rebuttal.

---

### Official Review · Reviewer_8oup · 2023-11-03

**Soundness:** 2 fair
**Presentation:** 2 fair
**Contribution:** 2 fair
**Rating:** 3
**Confidence:** 5

**Summary:**

The author proposes a distillation framework to mutually transfer both spatial and temporal knowledge from a teacher model to smaller student model.

**Strengths:**

The paper is easy to follow and presents the goal of the proposed method clearly.

**Weaknesses:**

The experimental section of the paper is very unclear. It is unclear what is meant to be the primary models compared against. Simply comparing distillation is not useful as sleep staging is not a task where it is very important. The processing is performed after completion. There is no necessity of real-time processing and complexity is not a big issue. Moreover there is no real analysis of model complexitySimply training a smaller network can be more beneficial than performing distillation even if we want to reduce complexity. There are other papers which use spatial-temporal relationships in sleep staging such as BSTT [1] and GraphSleepNet [2]. At a minimum this process should be compared with these methods. The results presented in Table 4 also does not make sense based on the relative positions of these positions in other papers. ISRUC-III has only 10 subjects so was the evaluation performed on only one subject? That is very unreliable.


[1] Liu, Y., & Jia, Z. (2022, September). Bstt: A bayesian spatial-temporal transformer for sleep staging. In The Eleventh International Conference on Learning Representations.
[2] Jia, Z., Lin, Y., Wang, J., Zhou, R., Ning, X., He, Y., & Zhao, Y. (2021, January). GraphSleepNet: adaptive spatial-temporal graph convolutional networks for sleep stage classification. In Proceedings of the Twenty-Ninth International Conference on International Joint Conferences on Artificial Intelligence (pp. 1324-1330).

**Questions:**

Please address the weaknesses

---

> ### Author Response · Authors · 2023-11-22
> **Response to Reviewer 8oup (1/2)**
>
> > Q1: It is unclear what is meant to be the primary models compared against.
>
> A1: The manuscript proposes a generic knowledge distillation framework that can be applied on most multi-channel sleep models. It is not model-specific. The teacher and student model we choose to apply knowledge distillation is only an example to show the effectiveness of the knowledge distillation framework. The comparison should be between our knowledge distillation framework and other knowledge distillation frameworks on the same model.
>
> Because it is a generic knowledge distillation framework for most multi-channel sleep models, it can be applied on other types of model structures. We add new experiments on CNN+GCN structures on ISRUC-III dataset. The teacher model reaches 85.93% accuracy and 83.95% F1-score. The results of KD methods applying on the corresponding student model are presented in section 5.3 Table 4:
>
> > | KD Methods                        | Accuracy | F1-score |
> > | --------------------------------- | -------- | -------- |
> > | Knowledge Distillation            | 75.07%   | 72.35%   |
> > | Decoupled Knowledge Distillation  | 82.44%   | 80.26%   |
> > | Fitnets                           | 81.88%   | 80.76%   |
> > | Neuron Selectivity Transfer       | 83.31%   | 80.94%   |
> > | Relational Knowledge Distillation | 76.68%   | 73.19%   |
> > | Distilling Knowledge from GCN     | 82.65%   | 79.69%   |
> > | Deep Mutual Learning              | 81.27%   | 77.84%   |
> > | Ours                              | 84.26%   | 81.22%   |
>
> > Q2: The processing is performed after completion. There is no necessity of real-time processing and complexity is not a big issue. Moreover there is no real analysis of model complexity.
>
> A2: To our best knowledge, the computational resources in medical scenarios are in short. Although real-time processing is not necessary in sleep stage classification, it still lacks of computational resources after completion. For example, in the embedded devices, data prefers to be processed locally. Most embedded devices for EEG acquisition do not have enough computing power to process local data. Even when uploaded to a server with sufficient computing power for processing, there are issues of data privacy and transmission costs, not to mention the fact that most hospitals cannot afford such servers. Therefore, it is necessary to compress the sleep models so that it can run quickly on embedded devices.
>
> We analyze the model complexity from aspects like numbers of parameters, FLOPs and model size in the experiment section 5.3 and table 2. The compression comes from the lightweight design of the student model.

---

> ### Author Response · Authors · 2023-11-22
> **Response to Reviewer 8oup (2/2)**
>
> > Q3: Simply training a smaller network can be more beneficial than performing distillation even if we want to reduce complexity.
>
> A3: It is true that simply training a smaller network can be a good way to achieve lightweight sleep stage classification. Applying knowledge distillation on existing sleep models is from anther prospective. Our proposed knowledge distillation framework is generic to most multi-channel sleep models. It doesn't need to design a new smaller structure from scratch but reduce the model scale from an existing model. It is also a useful way to achieve lightweight sleep stage classification. Moreover, our framework can also be used on the smaller network you mentioned and make it even smaller.  The smaller network can possess knowledge of the original model under some circumstances.
>
> > Q4: There are other papers which use spatial-temporal relationships in sleep staging such as BSTT [1] and GraphSleepNet [2]. At a minimum this process should be compared with these methods.
> >
> > [1] Liu, Y., & Jia, Z. (2022, September). Bstt: A bayesian spatial-temporal transformer for sleep staging. In The Eleventh International Conference on Learning Representations.
> >
> > [2] Jia, Z., Lin, Y., Wang, J., Zhou, R., Ning, X., He, Y., & Zhao, Y. (2021, January). GraphSleepNet: adaptive spatial-temporal graph convolutional networks for sleep stage classification. In Proceedings of the Twenty-Ninth International Conference on International Joint Conferences on Artificial Intelligence (pp. 1324-1330).
>
> A4: These two models are indeed representative and powerful examples in sleep stage classification area. However, the framework we proposed is generic knowledge distillation framework for most multi-channel sleep models which is not a sleep model but a framework applied on sleep models. So, it cannot be compared with BSTT [1] and GraphSleepNet [2].  Moreover, BSTT and GraphSleepNet can be used in our knowledge framework. We conduct experiments on teacher-student models based on the CNN+GCN structure, which is also the structure of GraphSleepNet. It also performs well by reducing the parameters by 60% while maintaining the performance with 1.7% loss of accuracy. Our framework can help applications of sleep models by greatly reducing the scale of multi-channel sleep models with different structures.
>
> > Q5: ISRUC-III has only 10 subjects so was the evaluation performed on only one subject? That is very unreliable.
>
> A5: We use the whole sleep period of the 10 subjects. It contains about 8,549 PSG sleep smaples. We also conduct experiments on a larger dataset called MASS-SS3. The results show that our method is still effective on the MASS-SS3. The details of the datasets are presented in section 5.1 Datasets:
>
> > ISRUC-III is obtained from a sample of 8,549 PSG sleeps over 8 hours from 10 healthy adult subjects, including one male and nine females. We use 8 subjects as the training set, 1 subject as the validation set, and 1 subject as the test set.
> >
> > MASS-SS3 contains 59,056 PSG sleep samples from the sleep data of 62 healthy subjects, including 28 males and 34 females. We also use 50 subjects as the training set, 6 subjects as the validation set, and 6 subjects as the test set.

---

### Author Response · Authors · 2023-11-22
**Responses to All Reviewer**

We thank all reviewers for their constructive feedback. Based on this feedback, we have made a number of changes to address the shortcomings of the paper.

In particular, our main updates are summarized as follow:

1. In 4.3 Mutual Distillation Framework, we change the order of equations and correct the mistakes in the Eq.(12-13)
2. In 5.2 experiment settings, we introduce a new architecture of CNN+GCN into experiments to further demonstrate the effectiveness of our knowledge distillation framework.
3. In 5.3 overall results, we add table 4 and update table 2 to present the results on the CNN+GCN architecture.
4. We add a new subsection 5.4 Visualization to show the visual analysis of spatial and temporal knowledge transfer.

We have further refined the Appendix which are summarized below:

1. In Appendix A.4, we add results on 6-channel EEG and 2-channel EOG to demonstrate the generality of oure framework on different channels.
2. In Appendix A.5, we add descriptions on the new CNN+GCN architectures and details of the model.
3. We add sensitivity analysis for loss weights in Appendix A.7,
4. We add experiments with different distance measurement functions in Appendix A.8
5. We move the clinical demand part in original 5.3 overall results to Appendix A.9.

---

### Meta-Review · Area_Chair_Waue · 2023-12-10

**Metareview:**

The paper explores a knowledge distillation approach for automatic sleep staging which is a classification task with 5 classes. Here a teacher network is used to supervised a student network as previously proposed in other contexts in ML. The method is evaluated on two datasets.

The contribution in terms of machine learning method is limited. Also as pointed out by some reviewers the experimental
evaluation is not ambitious (2 datasets to be compared with published works that have used thousands for nights to train SoTA models).

Despite having some practical relevance for sleep medicine as pointed by one enthusiast reviewer, the contribution is overall considered limited by the rest of reviewers and the AC. For these reasons, the paper cannot therefore be endorsed for publication at ICLR this year.

**Justification For Why Not Higher Score:**

Limited experiments and moderate ML contribution.

**Justification For Why Not Lower Score:**

The paper is clear with no clear flaw and it can have a practical relevance for sleep medicine.

---

### Decision · Program_Chairs · 2024-01-16

Reject